# Effects of non-pharmacological interventions on patients with sarcopenic obesity: A meta-analysis

**Jiajia Xu, Qingqing Hu, Jiaying Li, Yixi Zhou, Ting Chu**  *

Nursing School, Zhejiang Chinese Medical University, Hangzhou, Zhejiang Province, People's Republic of China

* chut@zcmu.edu.cn

## Abstract

### Objective

Systematic evaluation of the improvement effect of non-pharmacological intervention on patients with sarcopenic obesity.

### Methods

Wanfang, VIP, China National Knowledge Infrastructure (CNKI), SinoMed, Web of Science, Cochrane Library, PubMed, and Embase databases were searched systematically for randomized controlled trials (RCTS) and experimental studies of non-pharmacological interventions on patients with sarcopenic obesity. The retrieval period was from the establishment of the databases to October 31, 2022. Meta-analysis was conducted using RevMan 5.4.

### Results

A total of 18 studies involving 1,109 patients were included. Meta-analysis results showed that non-pharmacological interventions improved patients' body weight [mean difference, MD = −2.74, 95% CI (−4.79, −0.70), $P$ = 0.009], body fat percentage [MD = −0.67, 95% CI (−0.96, −0.38), $P$ < 0.00001], grip strength [MD = 1.29, 95% CI (0.81, 1.77), $P$ < 0.00001], gait speed [MD = 0.05, 95% CI (0.03, 0.07), $P$ < 0.00001], and knee extension strength [MD = 2.56, 95% CI (1.30, 3.82), $P$ < 0.0001].

### Conclusions

Non-pharmacological interventions can effectively improve the clinical symptoms and signs of patients with sarcopenic obesity. Dissemination of this information will be therapeutically useful.

### Trial registration

**Registration**. The PROSPERO No. is CRD42023403341.

**Data Availability Statement:** All relevant data are within the manuscript and its Supporting information files.

**Funding:** The authors received no specific funding for this work.

**Competing interests:** The authors have declared that no competing interests exist.

## Introduction

In 2022 the European Society of Clinical Nutrition and Metabolism (ESPEN), and the European Association for the Study of Obesity (EASO) issue recommendations to define Sarcopenic obesity (SO) as a pathological condition characterized by the coexistence of excessive obesity and loss of muscle mass or muscle function [1]. SO is particularly prevalent among the elderly population, and the population aged 65 and over represents 13% of the global population and will reach 2.1 billion in 2050 [2]. Health and nutrition surveys show that the prevalence of SO is 12.6% in men and 33.5% in women [3]. Considering the rapid growth of the global elderly population, it is estimated that in 2051, SO will affect 100–200 million people worldwide [4]. Studies have shown that SO significantly reduces the quality of life and increases the risk of falls, fractures, and disability compared to patients with reduced skeletal muscle alone or who are obese [5], and increases the risk of death [6]. A health and aging study suggests that during a 7-year follow-up period, patients with SO had a tripled risk of weakness and a 1.5-fold increased risk of disability [7]. Moreover, SO is related to the occurrence and deterioration of depression, cancer and dementia [8–10].

At present, there is no specific drug to treat sarcopenic obesity, and non-drug intervention is considered as the optimal approach [11]. The literature on non-drug intervention in patients with SO focuses on the use and effects of exercise, nutrition, psychological support, education, and self-management [12,13]. Studies have shown that planned physical exercise and nutritional supplements (such as whey proteins and amino acids) can improve muscle function [14], and exercise and nutrition programs help reduce the incidence of obesity [15,16]. Exercise can postpone the initiation and advancement of sarcopenic obesity by influencing protein metabolism, inflammation, regulation of mitochondrial mass, and enhance the release of myokines from diverse tissues into the circulatory system [8]. Nutritional intervention mainly includes calorie restriction and protein and vitamin D supplementation. Calorie restriction can reduce the detrimental effects of obesity on skeletal muscle by controlling weight, while protein and vitamin D supplementation can help to enhance muscle strength [14]. In addition, augmented protein intake stimulates muscle protein synthesis, thereby averting the loss of skeletal muscle mass caused by weight reduction [15].

Many studies have been conducted to investigate non-pharmacological interventions for sarcopenic obesity. For example, three systematic reviews have performed to evaluating the effects of three distinct exercise modalities, whole-body electrical stimulation, and protein supplementation in patients with sarcopenic obesity [16–18]. However, the effect of exercise and nutrition on obese individuals with sarcopenia is uncertain because several studies suggest that non-pharmacological interventions may not significantly improve the health status of patients with sarcopenic obesity [19,20]. Hence, in order to solve the above doubts and further verify the therapeutic effect of non-pharmacological interventions on patients with sarcopenic obesity, we carried out this meta-analysis. However, given the limited literature available and the feasibility of conducting a meta-analysis, this work focuses on the effects of exercise and nutritional interventions on patients with sarcopenic obesity.

## Material and methods

This meta-analysis fallowed the Preferred Reporting Items for Systematic Reviews and Meta-Analyses (PRISMA) statement guidelines [21], and is registered in the PROSPERO (CRD42023403341).

## Search strategy

A systematic search was conducted using the following databases: China National Knowledge Infrastructure (CNKI), Wanfang Data, VIP Database for Chinese Technical Periodicals, China Biomedical Literature Database, Web of Science, Cochrane Library, PubMed, and Embase. The search time frame was from the inception of each database to October 31, 2022, and manually searched for additional references to supplement the existing literature. The search strategy used a combination of subject headings and free-text terms (S1 File).

## Eligibility criteria

The inclusion criteria were based on the "PICOS" (population, intervention, comparison, outcome, and study) approach:

P: Patients diagnosed with sarcopenic obesity according to the most recent definition proposed by ESPEN and EASO in 2022 undergo a two-stage diagnostic process [1]. The screening stage involves evaluating the patient's body mass index (BMI) or waist circumference and screening for sarcopenia. Diagnosis is established when both obesity and sarcopenia criteria are met. The diagnostic stage focuses on assessing the patient's skeletal muscle functional parameters, such as muscle strength and physical function, as well as changes in body composition. The diagnosis of SO is confirmed when three specific indicators are met, and the diagnostic criteria for each indicator are based on race-specific reference ranges.

I: Non-pharmacological interventions, such as physical activity and nutritional interventions.

C: Comparison of therapeutic effects between non-pharmacological intervention (physical activity or nutritional intervention) group and a control group receiving no intervention or placebo.

O: Outcome indicators including body weight, body fat percentage, body mass index, grip strength, gait speed, knee extension strength.

S: Study type only including randomized controlled trials.

Exclusion criteria were (1) duplicate publication; (2) lack of access to full text; (3) articles in which the data for extraction were incomplete or unavailable.

## Selection process

Literature screening was performed independently by two researchers. Duplicate studies were removed using EndNote, followed by an initial assessment of titles and abstracts to identify articles that potentially meet the inclusion criteria. Subsequently, the full text of selected articles was thoroughly examined to determine their eligibility for inclusion, based on predefined inclusion and exclusion criteria. In the event of any discrepancies, a third researcher conducted an independent assessment to resolve disagreements.

## Data collection process

Data extraction for each study was independently conducted by both researchers, adhering to standardized criteria. The extraction process encompassed the following information: authors of the included literature, country of origin, publication timeframe, literature type, sample size, patient gender, intervention details, duration of intervention, and outcome measures. In the event of any discrepancies, a third researcher performed an independent assessment to resolve any disagreements.

## Methodological quality assessment

The randomized controlled trials included in this study were evaluated independently by two researchers according to the Cochrane 5.1.0 Handbook's risk of bias assessment criteria [22]. In the event of disagreements, a third researcher evaluated the studies. The evaluation criteria included selection bias, implementation bias, measurement bias, attrition bias, and reporting bias. If the studies met all the criteria, the likelihood of bias was low, and the quality of the literature was rated as A. If the studies partially met the criteria, the likelihood of bias was moderate, and the quality of the literature was rated as B. If the studies did not meet any of the criteria, the likelihood of bias was high, studies were rated as C and were excluded from this study.

## Synthesis methods

Statistical analysis was performed using RevMan 5.4. Mean and standard deviation (SD) of baseline and final measurements were extracted for the intervention and control groups, and the mean difference (MD) and SD of the change from baseline to final measurement were calculated. If SD was not reported, it was estimated using the formula provided in the Cochrane 5.1.0 handbook, which is SD = $\sqrt{N}$ × (upper limit of confidence interval − lower limit of confidence interval)/4.128 for each group. Transformed variables were entered into the statistical software using the same unit of measurement. The MD with a 95% confidence interval (CI) was used as the effect size.

## Reporting bias

The heterogeneity of the data was assessed using $I^2$ statistic. If $P > 0.1$ and $I^2 < 50\%$, a fixed-effects model was used for the meta-analysis. If $P < 0.1$ and $I^2 \geq 50\%$, clinical heterogeneity was assessed to determine whether a random-effects model should be used, followed by sensitivity analysis. A significance level of $P < 0.05$ was used to indicate statistical significance. Publication bias was assessed using a funnel plot.

# Results

## Study selection

A total of 2605 articles were identified through initial searches. After removing duplicates using Endnote 20 software and reviewing titles and abstracts, 33 articles were included for full-text review. Ultimately, 18 articles [23–40] were included in the analysis after quality assessment. The selection process is illustrated in Fig 1.

## Study characteristics

The basic characteristics of the 18 included studies are presented in Table 1 [23–40]. The studies were sourced from 6 countries, with 17 in English and 1 in Chinese. The publication years ranged from 2014 to 2022. The study population consisted of individuals with SO with or without non-pharmacological interventions.

## Description of interventions

**Exercise intervention protocols.** Eight studies investigated the effects of exercise intervention on patients with SO [23,24,28–30,32,34,37]. Two studies utilized three distinct exercise modalities, namely aerobic, resistance, and mixed exercise [23,29]. Among them, in one study, the intervention duration was set at 20 minutes with a moderate exercise intensity [23]. In the

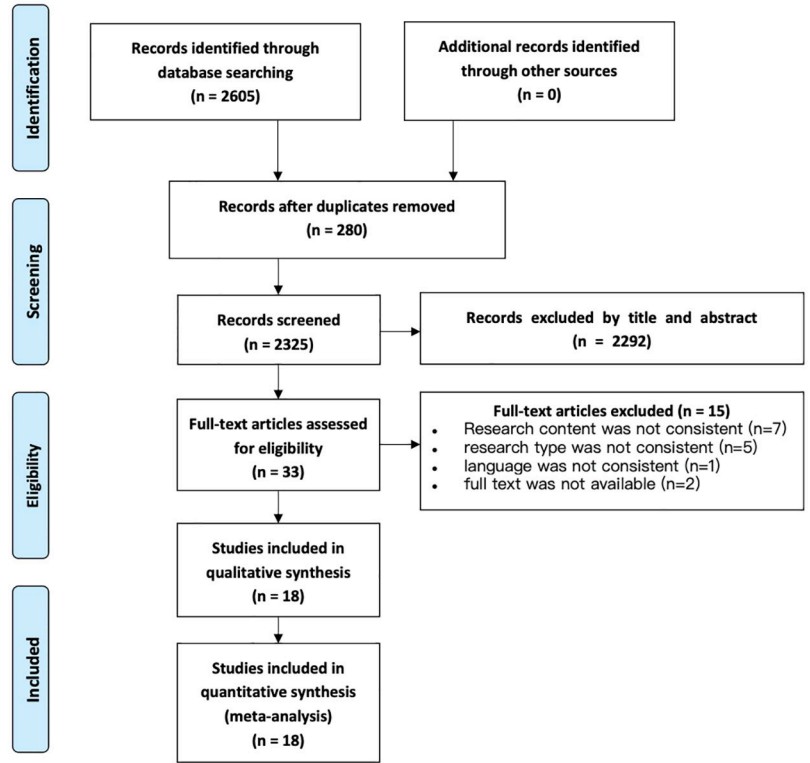

**Fig 1. Flowchart of studies included in this meta-analysis.**

other study, the intervention duration was 60 minutes with an exercise intensity of 60–70% of one repetition maximum (1RM) [29]. Additionally, the intervention frequency in both studies was twice a week over a period of 8 weeks. Meanwhile, five studies implemented resistance exercise interventions [28,30,32,34,37], One study employed equipment to facilitate both open chain and closed kinetic chain exercises [28]. The intervention duration was 60 minutes per session, with an exercise intensity based on 1RM. The intervention was conducted twice a week for a duration of 10 weeks. On the other hand, the remaining four studies utilized elastic bands for resistance exercises [30,32,34,37]. The intervention intensity was rated at 13 points on the Borg Scale, while the exercise duration varied from 35 to 60 minutes. The intervention frequency ranged from 2 to 3 times per week, with a total intervention duration spanning from 10 to 16 weeks. Additionally, there is a study that implemented high-speed circuit training [24] which was performed on 5 lower body and 6 upper body pneumatic exercise machines [24], performed 3 sets of 10–12 repetitions on each machine, 40–45 min per time, intensities ranging from 30 to 90% of 1RM, The intervention was administered with a frequency of twice per week over a period of 15 weeks.

## Nutrition intervention protocols

Two studies investigated the effects of nutrition intervention on patients with SO [27,33]. One study followed a high-protein diet with 1.2 g / kg body weight reference/day for 3 months [27], the other study used a hypocaloric high-protein diet with 1.2–1.4 g / kg body weight reference/ day for 4 months [33].

**Table 1. Characteristics of included studies.**

| Study | Country | Study design | Sample | | Intervening measure | | Duration | Outcome indicator |
|---|---|---|---|---|---|---|---|---|
| | | | EG | CG | EG | CG | | |
| Balachandran 2014 [24] | America | RCT | 11 | 10 | HSC: Training was performed on 5 lower body and 6 upper body pneumatic exercise machines 40–45 min per time | SH: Conventional strength/ hypertrophy, 55–60 min per time | 15 weeks, 2 times per week | BF%, GS$_1$ |
| Kemmler 2016 [25] | Germany | RCT | WB-EMS +P:25 WB-EMS:25 | 25 | WB-EMS: participants simultaneously performed a video guided WB-EMS program in a supine sitting/lying position with slight movements of the lower and upper limbs Protein supplementation: 40 g/day Vitamin D supplementation: 800 IU/day | dietary counseling, vitamin D supplementation: 800 IU/day | 26 weeks, 1 time per week | BF%, GS$_1$, GS |
| Kim 2016 [26] | Japan | RCT | EX+N: 36 Ex: 35 N: 34 | 34 | EX: Resistance, weight-bearing exercise, Aerobic training, 60 min per time N: Amino acid supplementation, 3 g of leucine enriched essential amino acid and 20 mg vitamin | Health education | 3 months, 2 times per week. | BF%, GS$_1$, GS, KES |
| Muscariello 2016 [27] | Italy | RCT | 54 | 50 | 1.2 g/kg DBW/day of proteins | 0.8 g/kg DBW/day of proteins | 3months | GS$_1$ |
| Vasconcelos 2016 [28] | Brazil | RCT | 14 | 14 | resistance exercise program, 60 min per time | Blank control | 10 weeks, 2 times per week. | GS |
| Chen 2017 [29] | Taiwan | RCT | RT: 22 AT: 24 CT: 25 | 22 | RT: weight-training equipment at 60–70% of one repetition maximum, 60 min per time AT: aerobic training in moderately intense, 60 min per time CT: each training mode once a week with the AT following 48 hours after the RT | maintained their day-to-day lifestyles and dietary habits and was prohibited from engaging in any exercises | 8 weeks, 2 times per week. | BW, BF%, GS$_1$, KES |
| Huang 2017 [30] | Taiwan | RCT | 18 | 17 | RT: perform individual resistance exercises by using elastic bands, 55min per time | health education booklet about SO and home exercise | 12 weeks, 3 times per week | BW, BF% |
| Kemmler 2017 [31] | Germany | RCT | WB-EMS +P:33 P: 33 | 34 | WB-EMS: bipolar electric (85Hz, 350μs, 4 s of strain to 4 s of rest), from 14 min to 20 min after 4 weeks. P: protein intake of 1.7–1.8 g/kg per day body mass. Cholecalciferol: 800 IU/day | Cholecalciferol: 800 IU/day | 16 weeks, 1.5 times per week | BF%, GS$_1$ |
| Liao 2017 [32] | Taiwan | RCT | RT: 25 | 21 | RT: Progressive RET using TheraBand products, 35–40 min per time | Blank control | 12 weeks, 3 times per week | BF%, GS$_1$, GS |
| Sammarco 2017 [33] | Italy | A pilot study | 9 | 9 | hypocaloric high-protein diet (1.2–1.4g/kg body weight reference/day) | Hypocaloric diet plus placebo | 4months | BW, GS$_1$ |
| Silva 2018 [34] | America | RCT | 8 | 41 | RT: three sets of 12–14 RM(1–4 weeks), 10–12 RM(5–8 weeks), 8–10 RM(9–12 weeks), 6–8 RM(13–16 weeks)40–50 minutes per time | Blank control | 16 weeks, 2 times per week | BW |
| Kemmler 2018a [35] | Germany | RCT | WB-EMS +P:33 | 34 | WB-EMS: 1.5 × 20 min/week (85 Hz, 350 μs, 4 s of strain–4 s of rest) with moderate-high intensity, 20 min per time P: whey protein powder supplements intake of 1.7–1.8 g/kg/body mass/d Cholecalciferol: 800 IU/day | Cholecalciferol: 800 IU/day | 16 weeks, 1.5 times per week | WC, BF% |
| Kemmler 2018b [36] | Germany | RCT | WB-EMS +P:33 P:33 | 34 | WB-EMS: bipolar electric(85 Hz,350 μs, 4 s of strain to 4 s of rest)20 min per time P: whey protein powder supplements of 1.7–1.8 g/kg/day body mass Cholecalciferol: 800 IU/day | Cholecalciferol: 800 IU/day | 16 weeks, 1.5 times per week | GS |

(*Continued*)

**Table 1.** (Continued)

| Study | Country | Study design | Sample | | Intervening measure | | Duration | Outcome indicator |
|---|---|---|---|---|---|---|---|---|
| | | | EG | CG | EG | CG | | |
| Liao 2018 [37] | Taiwan | RCT | EXP:33 | 23 | RT: Progressive RET using TheraBand products,50-55min per time | receiving no exercise intervention | 12 weeks, 3 times per week | BF%, GS |
| Zhou 2018 [38] | China | RCT | 23 | 25 | electric stimulator: stimulated with an electric stimulator for(5 Hz, 1 ms, 1.5 mA), 20 min per time received 20g essential amino acids orally The total calories of the food were 1.58 × (13.5 × weight (kg) + 487) | received 20g essential amino acids orally, twice per day for 28 weeks | 12 weeks, 1 times every 3days | BF% |
| Nabuco 2019 [39] | Brazil | RCT | 13 | 13 | RT: Resistance training program, after each training session, participants took 35g whey protein | placebo | 12 weeks, 3 times per week | WC |
| Wang 2019 [23] | China | RCT | RT: 20 AT: 20 CT: 20 | 20 | RT: resistance training, 20min per time AT: dynamic aerobic training, 20min per time CT: RT for 10 min, followed by AT for 20 min | Blank control | 8 weeks, 2 times per week | BW, BF%, GS₁, KES |
| Camajani 2022 [40] | Italy | A pilot study | 12 | 12 | VLCKD: 780–800 kcal/d IT: Interval Training, 30–35 min per time | VLCKD: 780–800 kcal/d | 6 weeks, 2 times per week | BMI, BW, WC |

EG experimental group; CG, control group; RCT, randomized controlled trial; RT, progressive resistance training; AT, aerobic exercise training; CT, muscle strength combined with aerobic exercise training; HSC, high-speed circuit; SH, Conventional strength/hypertrophy; WB-EMS+P, whole-body electromyostimulation with dietary supplement; WB-EMS, whole-body electromyostimulation; EX, exercise; N, nutrition; VLCKD+IT, Very Low Calorie Ketogenic Diet + interval training; VLCKD, Very Low Calorie Ketogenic Diet; BW, body weight; BF%, body fat percentage; GS₁, grip strength; KES, knee extensive strength; GS, gait speed; WC, waist circumstance.

## Exercise combined nutrition intervention protocols

Eight studies researched the effect of exercise combined nutrition intervention on SO patients [25,28,31,35,38,36–40]. Among them, Four studies involved WB-EMS in combination with protein supplementations [25,31,35,36], The WB-EMS was applied with a frequency of 85 Hz and an impulse width of 350 μs intermittently with 4–6 s of EMS simulation using a direct impulse boost and 4 s of rest, and the intensity of electrical stimulation ranged from moderate to high. The nutrition supplements were 500 mg calcium / day and 800IU cholecalciferol, protein intake of 1.7–1.8 g/kg per day body mass. Two studies employed a combined intervention consisting of resistance exercise and nutritional supplementation [26,39]. Patients engaging in resistance exercise, after each training session, participants took 35 g whey protein or took 3 g of leucine enriched essential amino acid and 20 mg vitamin. Additionally, one study implemented a very low ketogenic diet in conjunction with nutritional intervention involving a very low ketogenic diet (780–800 kcal/day) took physical exercise for 30–35 min each session [40]. while another study utilized a combined intervention of acupuncture and supplementation [38]. Participants received electric stimulator for 20 min with a frequency of 5 Hz, wave duration of 1 ms, and strength of 1.5 mA, once every 3 days for 12 weeks, and took essential amino acids orally twice per day (20 g in total) for 28 weeks.

## Study risk-of-bias assessment

In this study, 18 articles [23–40] were included, of which 8 articles [25,28,30–32,35–37] were rated as high quality and 10 articles [23,24,26,27,29,33,34,38–40] were rated as medium

quality. Each article [23–40] formulated inclusion and exclusion criteria for the study subjects. Fourteen articles [23–28,30–32,35–39] described the methods and process of randomization, 11 articles [23,25,28,30–31,35–37,39,40] mentioned allocation concealment, and 11 articles [24,25,27–32,35–37] used blinding for outcome assessors. Seventeen articles [23–26,28–40] reported on the loss to follow-up or used intention-to-treat analysis to reduce the impact of missing data on intervention outcomes. All articles [23–40] compared the baseline characteristics of the study subjects, such as age, gender, and disease. Due to the difficulty of blinding intervention providers and study subjects for exercise and nutrition interventions, 6 articles [23,26,29,33,34,40] did not describe the implementation of blinding. However, the outcome indicators of the meta-analysis were objective, and the impact of not implementing blinding on the study results was minimal. The presence of reporting bias remained uncertain in two articles [28,34], while no reporting bias was observed in the remaining 16 articles. The quality assessment of the articles is shown in S1 Fig. (Risk of bias graph) and S2 Fig. (Risk of bias summary).

## Effects of non-pharmacological intervention on body weight and body fat percentage

Six studies [23,28,39,33,34,40] compared the effects of non-pharmacological interventions on weight, with no heterogeneity observed among the studies ($I^2$ = 0%, P = 0.83) and a fixed-effects model was used for analysis. The results showed that non-pharmacological interventions could improve patients' weight [MD = −2.74, 95% CI (−4.79, −0.70), P = 0.009].

Subgroup analysis was performed according to the type of non-pharmacological intervention. The results showed that exercise intervention could improve patients' weight [MD = −2.99, 95% CI (−5.16, −0.83), P = 0.007], while there was no statistically significant effect of nutritional intervention, or the combined intervention of exercise and nutrition, on patients' weight [MD = 0.2, 95% CI (−11.01, 11.41), P = 0.97; MD = −1.00, 95% CI (−8.68, 6.68), P = 0.80], (Fig 2).

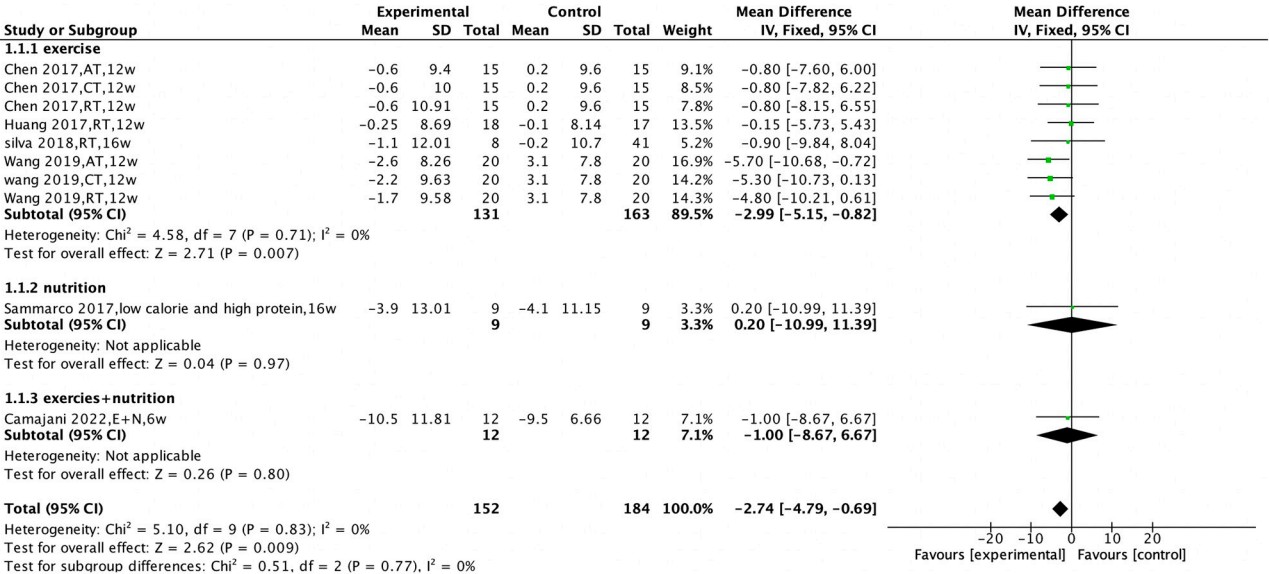

**Fig 2. Effect of non-pharmacological interventions on the body weight of sarcopenic obesity patients.**

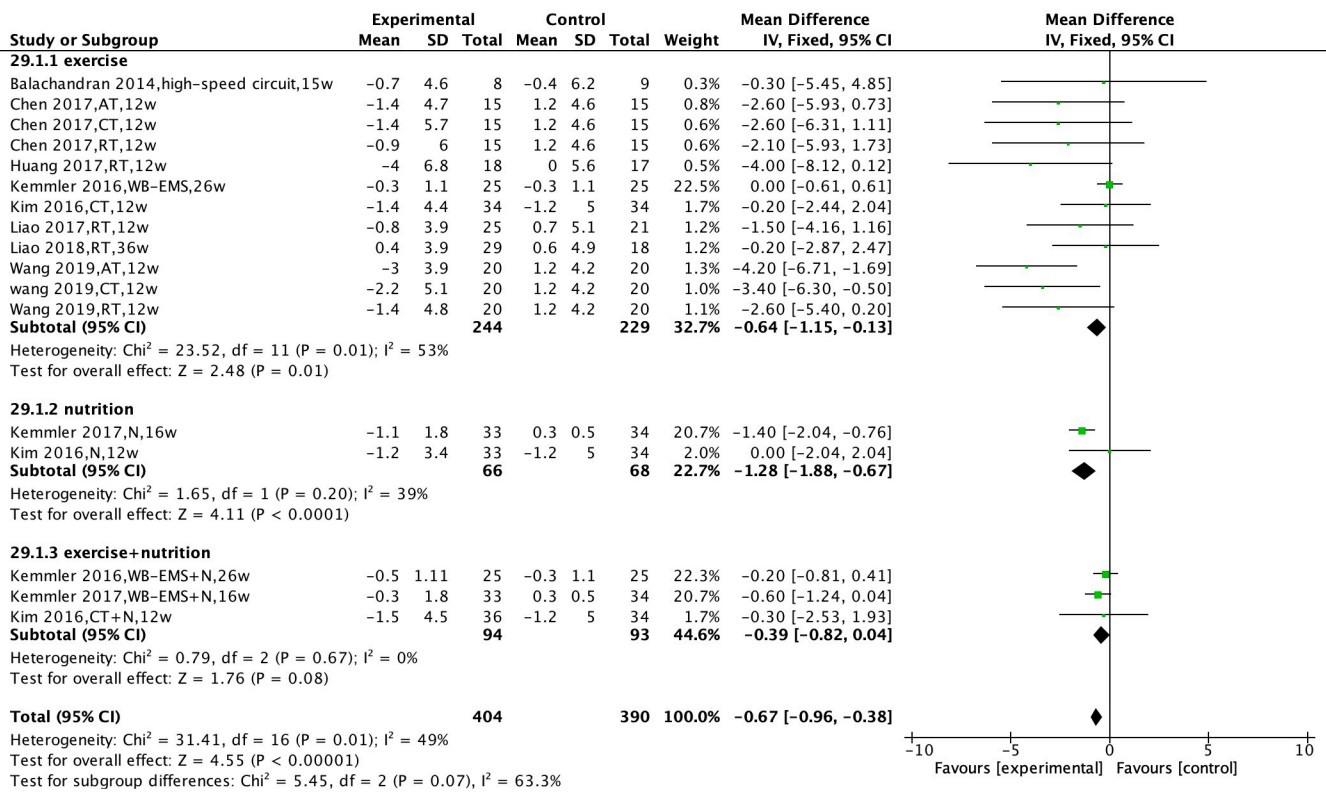

**Fig 3. Effect of non-pharmacological interventions on the body fat percentage of sarcopenic obesity patients.**

Nine studies [23–26,29–22,37] were included to compare the effects of non-pharmacological interventions on body fat percentage. There was no significant heterogeneity between the studies ($I^2$ = 49%, P = 0.01), and a fixed-effects model was used for analysis. The results showed that non-pharmacological interventions could improve the body fat percentage of patients [MD = −0.67, 95% CI (−0.96, −0.38), P < 0.00001].

Subgroup analysis was performed based on the type of non-pharmacological intervention. The results showed that exercise intervention and nutritional intervention could separately reduce the body fat percentage of patients [MD = −0.64, 95% CI (−1.15, −0.13), P = 0.01, MD = −1.28, 95% CI (−1.88, −0.67), P < 0.0001], while the combined intervention of exercise and nutrition had no statistically significant effect on the body fat percentage [MD = −0.39, 95% CI (−0.82, 0.04), P = 0.08] (Fig 3).

### Effects of non-pharmacological interventions on grip strength, gait speed, and knee extensor strength

Nine studies [23–27,29,31–33] compared the effect of non-pharmacological interventions on grip strength. There was no heterogeneity among studies ($I^2$ = 0%, P = 0.68), and a fixed-effects model was used for analysis. The results showed that non-pharmacological interventions improved grip strength [MD = 1.29, 95% CI (0.81, 1.77), P<0.00001].

Subgroup analysis was performed according to the type of non-pharmacological interventions. The results showed that exercise intervention, nutritional intervention, and exercise combined with nutritional intervention improved grip strength [MD = 0.96, 95% CI (0.16,

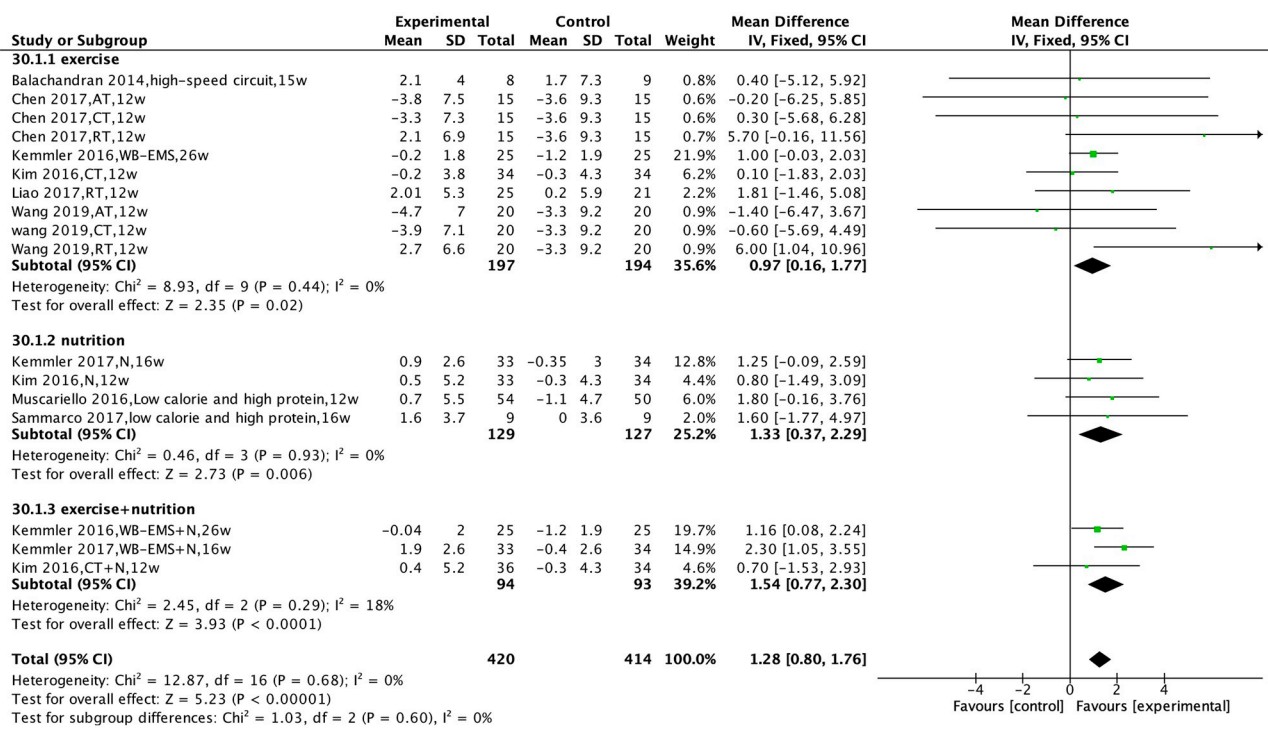

**Fig 4. Effect of non-pharmacological interventions on the grip strength of sarcopenic obesity patients.**

1.77), P = 0.02; MD = 1.36, 95% CI (0.40, 2.31), P = 0.005; MD = 1.54, 95% CI (0.77, 2.30), P<0.0001] (Fig 4).

Six studies [25,26,28,32,36,37] compared the effects of non-pharmacological interventions on gait speed. There was no significant heterogeneity among the studies ($I^2$ = 25%, P = 0.23), and a fixed-effects model was used for analysis. The results showed that non-pharmacological interventions could improve gait speed [MD = 0.05, 95% CI (0.03, 0.07), P<0.00001].

Subgroup analysis was performed based on the type of non-pharmacological intervention. The results showed that exercise intervention, nutritional intervention, and combined exercise and nutritional intervention could improve gait speed [MD = 0.07, 95% CI (0.01, 0.12), P = 0.01, MD = 0.10, 95% CI (0.00, 0.20), P = 0.04, MD = 0.05, 95% CI (0.03, 0.07), P<0.00001] (Fig 5).

Four studies [23,26,28,39] compared the effect of non-pharmacological interventions on knee extensor strength. There was no heterogeneity among the studies ($I^2$ = 0%, P = 0.55), and a fixed-effects model was used for analysis. The results showed that non-pharmacological interventions improved knee extensor strength [MD = 2.56, 95% CI (1.30, 3.82), P<0.0001].

Subgroup analysis was performed based on the type of non-pharmacological intervention. The results showed that exercise interventions improved knee extensor strength [MD = 3.28, 95% CI (1.66, 4.90), P<0.0001], while there was no statistically significant effect of nutrition intervention, or the combined intervention of exercise and nutrition, on knee extensor strength [MD = 0.90, 95% CI (−1.97, 3.77), P = 0.54; MD = 2.00, 95% CI (−0.77, 4.78), P = 0.16] (Fig 6).

## Subgroup analysis

Subgroup analysis was conducted based on the duration of non-pharmacological interventions, with durations of 1–10 weeks and 11–36 weeks. The results showed no significant

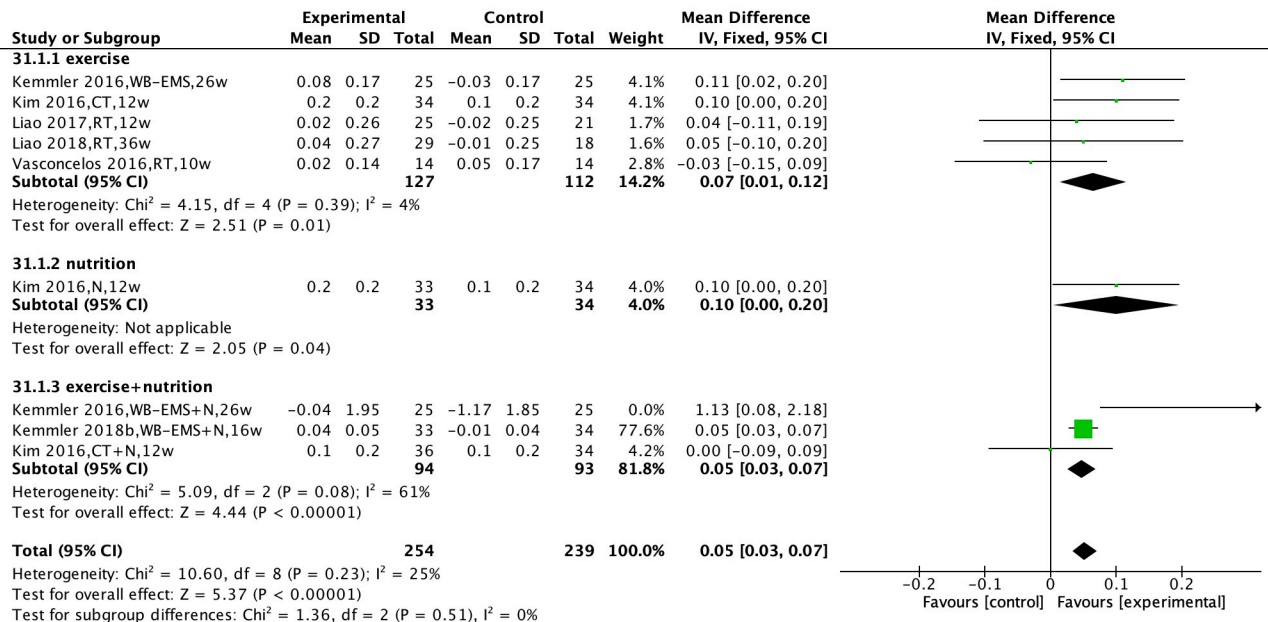

**Fig 5. Effect of non-pharmacological interventions on the gait speed of sarcopenic obesity patients.**

differences in body weight [$Chi^2$ = 3.95, P = 0.05, $I^2$ = 74.7%], body fat percentage [$Chi^2$ = 0.34, P = 0.56, $I^2$ = 0%], grip strength [$Chi^2$ = 0.01, P = 0.91, $I^2$ = 0%], gait speed [$Chi^2$ = 1.68, P = 0.20, $I^2$ = 40.3%], and knee extension strength [$Chi^2$ = 0.25, P = 0.62, $I^2$ = 0%] between the two intervention durations (See S2 File for details).

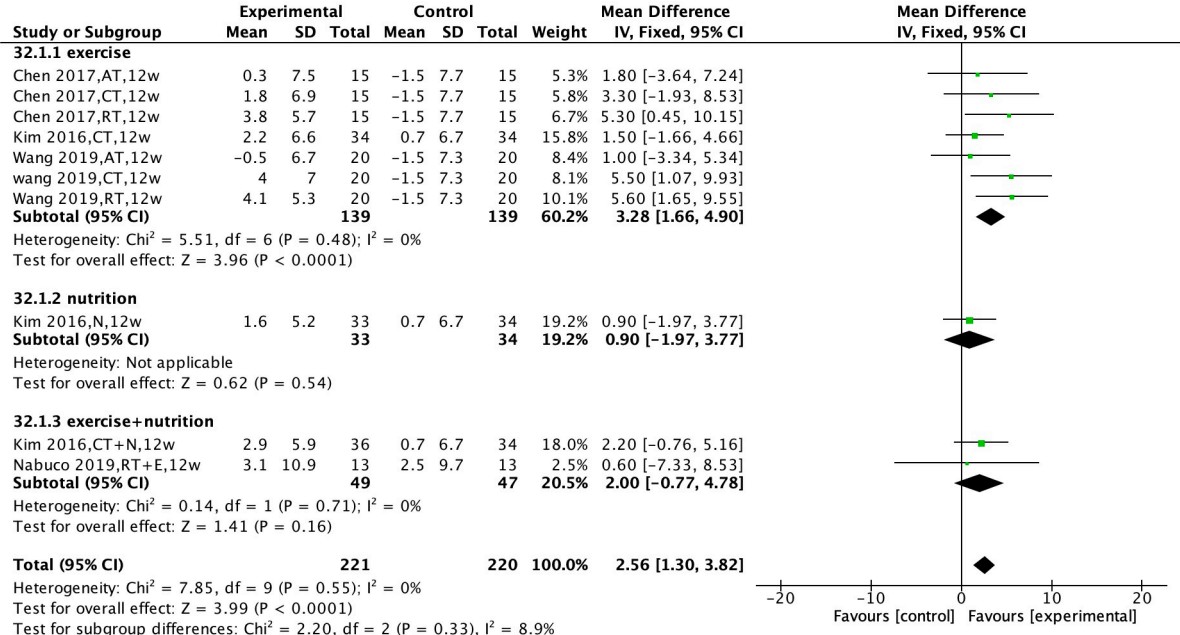

**Fig 6. Effect of non-pharmacological interventions on the knee extensive strength of sarcopenic obesity patients.**

## Publication bias

Studies with body weight, body fat percentage, grip strength, and knee extension strength as outcome measures were found to have publication bias (Figs 7–10).

## Discussion

In this meta-analysis, a total of 18 studies were included to compare the effects of exercise interventions, nutritional interventions, and combined exercise with nutritional interventions

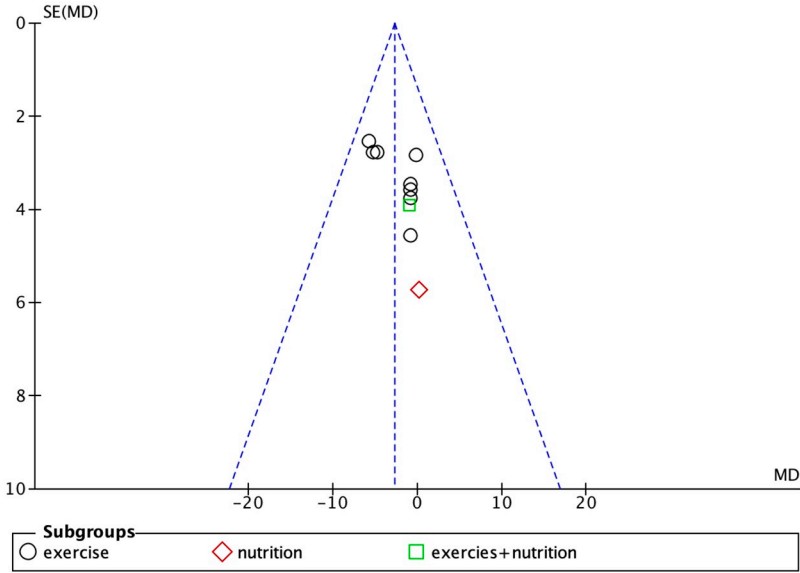

**Fig 7. Funnel plot for body weight.**

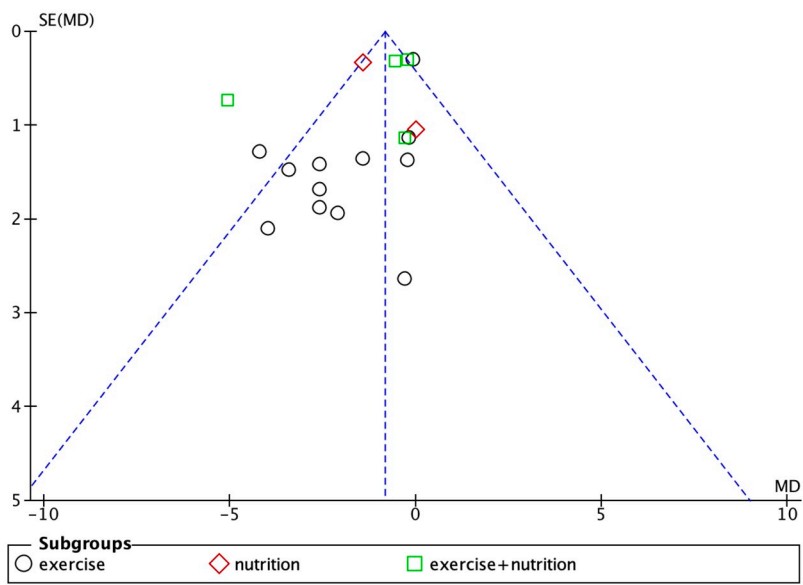

**Fig 8. Funnel plot for body fat percentage.**

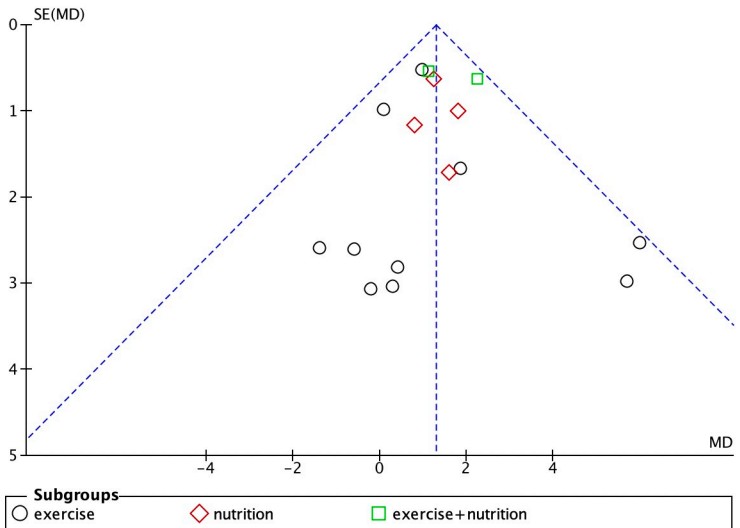

**Fig 9. Funnel plot for grip strength.**

on body composition and physical performance in elderly patients with sarcopenic obesity. The findings revealed that exercise interventions led to significant reductions in body weight and body fat percentage, along with improvements in grip strength, gait speed, and knee extension muscle strength. Nutritional interventions were found to enhance body fat percentage, grip strength, and gait speed, while demonstrating no significant effect on body weight and knee extension muscle strength. Furthermore, the combination of exercise and nutritional intervention showed improvements in grip strength and step speed, with no significant effects observed on body weight, body fat percentage, and knee extensor strength.

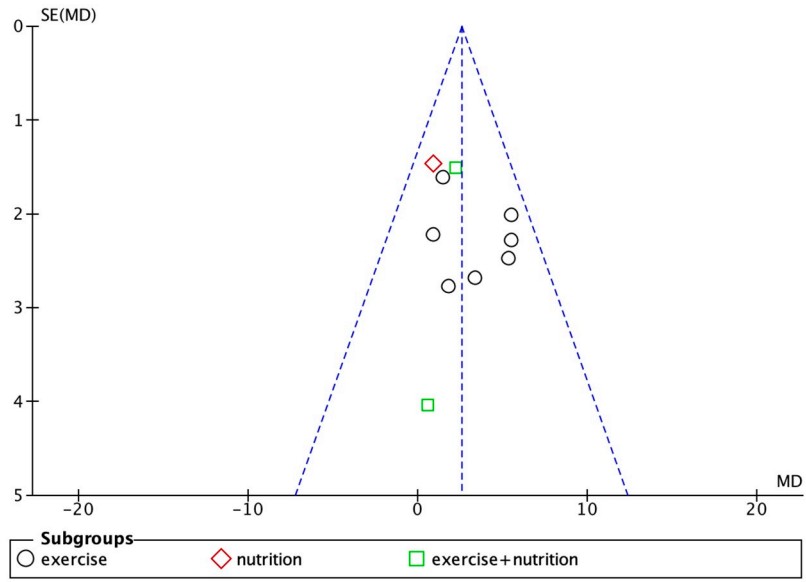

**Fig 10. Funnel plot for knee extensive strength.**

The findings of this study demonstrated a significant improvement in physical performance and body composition following exercise intervention, which is consistent with the findings reported by Hita-Contreras [41]. Garatachea et al [42] indicates that exercise can enhance physical performance in older adults through the regulation of gene expression, hormone levels (such as testosterone and IGF-1), and metabolic functions. In addition, Yin et al [43] reports that exercise interventions exhibited more consistent and favorable effects in reducing body fat and increasing muscle strength compared to nutritional interventions. Therefore, the evidence above shows that exercise has a certain effect on the treatment of SO and helps to prevent and control SO.

Chen et al [29] carried out resistance exercise with an intensity equivalent to 60–70% of a maximum repetition (1RM), while Liao et al [32] and Huang et al [30] adopted the subjective sensory exercise load assessment scale, and obtained an intensity grade of 13 points on the scale. For the frequency and duration of exercise, experts suggest that the elderly should do at least 150 minutes of moderate and high-intensity aerobic exercise every week, and at the same time do resistance training twice a week [44]. Some researchers suggest that the frequency of exercise for obese patients with sarcopenia can be 2 to 3 times a week [45] and the duration of exercise can be 20 to 60 minutes [32]. According to environmental factors and the health status of the subjects, the duration of intervention is usually limited to 8 to 16 weeks, and the duration of most studies is between 12 and 16 weeks [37]. The above results show that the existing exercise interventions for patients with sarcopenic obesity include aerobic exercise, resistance exercise, the combination of the two and whole body electrical stimulation, and the intensity of exercise is equal to or higher than the medium level.

Nutritional interventions include micronutrient supplementation, calorie restriction and protein supplementation. Findings of Cheng [46] and Liao et al [47] indicate that nutritional intervention can reduce weight, improve grip strength and walking speed. It is reported that vitamin D deficiency can lead to decreased muscle strength and increase the risk of falls and fractures [48]. The American Geriatrics Association recommends that people aged 65 and above take 1000 IU of vitamin D3 and calcium every day to maintain the serum vitamin D level at 30 ng/mL or above, and a daily caloric intake between 500 to 1000 kilocalories can reduce body weight by 8% to 10% within 6 months [49]. The recommended intake of protein is 0.8g/kg/ day, but the intake of protein should not be lower than 1.6g/kg/ day during high-intensity resistance training [50]. However, in a meta-analysis of the effects of low-calorie and high-protein diets and low-calorie and low-protein diets on the grip strength of obese patients with sarcopenia, no statistically significant difference was observed between the two groups [43], which may be because the duration of nutrition intervention is relatively short (12 weeks) in the included studies, which affects the full presentation of nutrition improvement effect. The above findings suggest that although the stability of nutritional intervention in reducing weight and improving knee extensor strength may not be as good as that of exercise intervention, the significance of nutritional intervention in enhancing body composition and muscle strength should not be ignored. Therefore, the effectiveness of nutritional intervention on patients with sarcopenic obesity is still a controversial topic, which needs further study.

The results of our meta-analysis showed that non-drug intervention can effectively improve the grip strength and gait speed of patients with sarcopenic obesity, which is consistent with the findings of Liao et al [47]. However, Hsu et al [51] reports that the combination of nutrition and exercise training had no obvious effect in the obese people with sarcopenia. The inconsistency of the results may be related to the differences in the duration of intervention, the type and intensity of exercise, and the frequency and quantity of nutritional supplements. The above reports suggest that the effects of exercise combined with nutrition on enhancing body composition and muscle strength are also inconsistent.

The results of our study showed that the duration of the intervention did not have a significant impact on the body fat percentage, limb skeletal muscle mass, grip strength, and gait speed of SO patients, which is consistent with the report by Hita-Contreras et al [41]. Muscariello et al [27] and Zhou et al [38] suggest that the effect of nutritional intervention may not be evident in the early stages and may require a longer period to manifest. Further exploration is needed on this issue.

## Limitations and future research

The limitations of this study are as follows: (1) the included studies used different definitions and diagnostic criteria for SO, which may affect the final assessment of the results; (2) some indicators have limited literature, which may lead to bias; (3) only Chinese- and English-language studies were included, and literature retrieval may be inadequate; (4) while this study provides suggestions on exercise and nutrition intervention, it does not establish a clear plan for non-drug intervention to treat sarcopenic obesity. Therefore, in future research, the following points need to be addressed: (1) it is necessary to establish and use a uniform definition and diagnostic criteria for SO; (2) efforts should be made to reduce bias and heterogeneity in the literature; (3) large-scale randomized controlled trials still serve as the best clinical trial method; (4) it is necessary to establish an evidence-based and clear non-drug intervention plan for sarcopenic obesity. Due to the limitations of this study, large-scale and high-quality randomized controlled trials are needed to further verify the results.

## Conclusion

Non-pharmacological interventions can effectively improve some clinical symptoms and signs of SO patients, which is worthy of promotion and use. When providing clinical guidance for exercise or nutritional interventions for elderly people with sarcopenic obesity, healthcare professionals should develop non-pharmacological intervention programs based on the individual health status of the subject. The lack of consistency in the effect of exercise intervention in different studies may be related to the differences in the definition and diagnostic criteria of sarcopenic obesity in different studies or the differences between different nationalities and races. The innovation of this meta-analysis is to test whether exercise and nutrition intervention is effective for sarcopenic obesity, and give a reasonable answer to the existing controversy.

## Supporting information

**S1 Checklist. PRISMA 2020 checklist.**
(DOCX)

**S1 Fig. Risk of bias graph.**
(TIF)

**S2 Fig. Risk of bias summary.**
(TIF)

**S1 File. Search strategy.**
(DOCX)

**S2 File. The effect of time-based intervention subgroup on sarcopenic obesity patients.**
(DOCX)

**S3 File. List of excluded 15 studies and reasons.**
(DOCX)

## Author Contributions

**Conceptualization:** Jiajia Xu, Ting Chu.

**Data curation:** Qingqing Hu, Jiaying Li, Yixi Zhou, Ting Chu.

**Formal analysis:** Jiajia Xu, Qingqing Hu, Jiaying Li, Yixi Zhou, Ting Chu.

**Methodology:** Jiajia Xu, Qingqing Hu, Jiaying Li, Ting Chu.

**Supervision:** Ting Chu.

**Writing – original draft:** Jiajia Xu.

**Writing – review & editing:** Ting Chu.

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
