## [Decision Letter · Decision Letter 0]

8 May 2023

PONE-D-23-08981Effects of non-pharmacological interventions on patients with sarcopenic obesity: A meta-analysisPLOS ONE

Dear Dr. Chu,

Thank you for submitting your manuscript to PLOS ONE. After careful consideration, we feel that it has merit but does not fully meet PLOS ONE’s publication criteria as it currently stands. Therefore, we invite you to submit a revised version of the manuscript that addresses the points raised during the review process.

We look forward to receiving your revised manuscript.

Kind regards,

Meisam Akhlaghdoust, M.D., M.P.H.

Academic Editor

PLOS ONE

Journal Requirements:

2. "Thank you for stating the following financial disclosure: 

Reviewers' comments:

Reviewer's Responses to Questions

**Comments to the Author**

1. Is the manuscript technically sound, and do the data support the conclusions?

Reviewer #1: Yes

Reviewer #2: Yes

2. Has the statistical analysis been performed appropriately and rigorously? 

Reviewer #1: No

Reviewer #2: Yes

3. Have the authors made all data underlying the findings in their manuscript fully available?

Reviewer #1: No

Reviewer #2: Yes

4. Is the manuscript presented in an intelligible fashion and written in standard English?

Reviewer #1: Yes

Reviewer #2: Yes

5. Review Comments to the Author

Reviewer #1: I am honored to be invited to review this manuscript. This study conducted a meta-analysis of the effects of non-pharmacological interventions on patients with sarcopenic obesity, and the results showed that non-pharmacological interventions can effectively improve some clinical symptoms and signs of sarcopenic obesity patients, which is worth promoting and using. However, some issues should be addressed.

1.The background seems very simple. The author should provide necessary information, such as the definition scope of non-pharmacological interventions. The entire article only describes exercise therapy and nutritional support, is it comprehensive? Is there any support from the original literature?

2. The inclusion criteria are inconsistent with the abstract description, should they be included in the experimental study?

3. The authors limited their search to title or abstract. For instance, “[Title/Abstract]” was used in PubMed. I am not sure that their search was comprehensive. Please explain why you limited the search to title of abstract. Moreover, other databases did not provide specific retrieval strategies.

4. The full text number format needs to be unified, and the title should be placed below the image.

5. Due to different types of exercise therapy and nutritional support, the benefits for patients with different severity levels also vary, making it difficult to measure the impact of a single method. How to solve this problem?

Reviewer #2: This article has some benefits about Effects of non-pharmacological interventions on patients with sarcopenic obesity even though it has some issues which should be improved. To sum up, it is proper for scientists and student know more about this subject.

6. PLOS authors have the option to publish the peer review history of their article (what does this mean?). If published, this will include your full peer review and any attached files.

Reviewer #1: No

Reviewer #2: No

---

## [Author Response · Author response to Decision Letter 0]

19 May 2023

Reviewer #1: 

I am honored to be invited to review this manuscript. This study conducted a meta-analysis of the effects of non-pharmacological interventions on patients with sarcopenic obesity, and the results showed that non-pharmacological interventions can effectively improve some clinical symptoms and signs of sarcopenic obesity patients, which is worth promoting and using. However, some issues should be addressed.

1.The background seems very simple. The author should provide necessary information, such as the definition scope of non-pharmacological interventions. The entire article only describes exercise therapy and nutritional support, is it comprehensive? Is there any support from the original literature?

Response: 

Thanks for your kindly help, we have revised the section of Introduction according to your suggestion. 

2. The inclusion criteria are inconsistent with the abstract description, should they be included in the experimental study?

Response: 

It has been revised, and the experimental study is added.

3. The authors limited their search to title or abstract. For instance, “[Title/Abstract]” was used in PubMed. I am not sure that their search was comprehensive. Please explain why you limited the search to title of abstract. Moreover, other databases did not provide specific retrieval strategies.

Response: 

The search strategy used a combination of subject headings and free-text terms (S1 File. Search strategy). Based on your advice, we use “All fields” to search related literature, however, the literature available for meta-analysis has not changed at all. 

4. The full text number format needs to be unified, and the title should be placed below the image. 

Response: 

It has revised. 

5. Due to different types of exercise therapy and nutritional support, the benefits for patients with different severity levels also vary, making it difficult to measure the impact of a single method. How to solve this problem?

Response: 

Good question. We think this problem can be solved by using normative intervention measures, comprehensive intervention, layered intervention, individualized intervention, Long-term tracking and evaluation of the effects. 

Reviewer #2: 

This article has some benefits about Effects of non-pharmacological interventions on patients with sarcopenic obesity even though it has some issues which should be improved. To sum up, it is proper for scientists and student know more about this subject.

Response: 

Thanks a lot for your support. We have revised it based on your suggestion.

---

## [Decision Letter · Decision Letter 1]

20 Jun 2023

PONE-D-23-08981R1Effects of non-pharmacological interventions on patients with sarcopenic obesity: A meta-analysisPLOS ONE

Dear Dr. Chu,

Thank you for submitting your manuscript to PLOS ONE. After careful consideration, we feel that it has merit but does not fully meet PLOS ONE’s publication criteria as it currently stands. Therefore, we invite you to submit a revised version of the manuscript that addresses the points raised during the review process.

We look forward to receiving your revised manuscript.

Kind regards,

Meisam Akhlaghdoust, M.D., M.P.H.

Academic Editor

PLOS ONE

Reviewers' comments:

Reviewer's Responses to Questions

**Comments to the Author**

1. If the authors have adequately addressed your comments raised in a previous round of review and you feel that this manuscript is now acceptable for publication, you may indicate that here to bypass the “Comments to the Author” section, enter your conflict of interest statement in the “Confidential to Editor” section, and submit your "Accept" recommendation.

Reviewer #1: All comments have been addressed

Reviewer #3: (No Response)

2. Is the manuscript technically sound, and do the data support the conclusions?

Reviewer #1: Yes

Reviewer #3: Partly

3. Has the statistical analysis been performed appropriately and rigorously? 

Reviewer #1: Yes

Reviewer #3: Yes

4. Have the authors made all data underlying the findings in their manuscript fully available?

Reviewer #1: Yes

Reviewer #3: Yes

5. Is the manuscript presented in an intelligible fashion and written in standard English?

Reviewer #1: Yes

Reviewer #3: Yes

6. Review Comments to the Author

Reviewer #1: The manuscript was conducted on the impact of non-pharmacological interventions on patients with sarcopenic obesity, indicating that non-pharmacological interventions can effectively improve some clinical symptoms and signs of patients of sarcopenic obesity patients, which is beneficial for clinical promotion and use. The author has made revisions one by one based on the relevant suggestions put forward by the editor. The article has a clear approach and the method is appropriate. It is recommended to publish it.

Reviewer #3: Thank you for inviting me to review the paper titled "Effects of Non-Pharmacological Interventions on Patients with Sarcopenic Obesity: A Meta-Analysis." It is an interesting review paper, but I have some concerns and suggestions for improvement.

Background

1. The background section needs stronger justifications to support the need for this systematic review and meta-analysis. There have already been several systematic reviews and meta-analyses on this topic, so it is important to address what other researchers have concluded in their papers, what research gaps this paper attempts to address, and why this review is necessary. Commenting on other similar systematic reviews to identify research gaps would strengthen the background section.

2. In lines 68-69, the authors state that there is limited literature on the feasibility of the interventions. However, I disagree, as several review papers on interventions for people with sarcopenic obesity have been published, such as “Yin YH, Liu JYW, Fan TM, Leung KM, Ng MW, Tsang TY, Wong KP, Välimäki M. Effectiveness of Nutritional Advice for Community-Dwelling Obese Older Adults With Frailty: A Systematic Review and Meta-Analysis. Front Nutr. 2021 Jun 29;8:619903. doi: 10.3389/fnut.2021.619903.” Thus, the authors should critique other similar systematic reviews to explain why the current review was considered necessary.

3. It is unclear whether this paper focuses on all types of non-drug interventions or only exercise and nutritional interventions. The authors should clarify this in the background section and align it with the title, paper selection criteria, and other parts of the manuscript.

4. The authors should briefly describe how the interventions (i.e., non-drug, exercise alone, nutrition alone, or combined, etc.) might work on people with sarcopenic obesity to provide a more comprehensive understanding.

Objective:

5. In line 72, the authors state that the meta-analysis aimed to "confirm" the effects of the interventions…... However, I feel that the word "confirm" is too strong. It would be better to use the word "explore" instead.

6. The objective should be revised using the PICO framework or one of its variants to clearly state the comparisons that were made. This will provide more clarity and specificity to the objective statement.

Methods - paper selection and data extraction process:

7. Please follow the headings and subheadings according to the PRISMA 2020 expanded checklist. For the Methods section, the headings should include eligibility criteria, information sources, search strategy, and selection process, among others, as specified by the checklist. This comment is also relevant to other sections of the manuscript.

8. The authors need to provide a better definition of targeted intervention(s) in the background section and align it with the eligibility criteria in this section.

9. As mentioned in the background section, sarcopenic obesity refers to the coexistence of obesity and sarcopenia. Therefore, the inclusion criteria should include both indicators to reflect this co-existing body structure. This should include at least one criterion for obesity and one for sarcopenia.

10. It should specify the year of publication for the studies included in this review paper.

11. It should report whether they conducted any hand searches in addition to their electronic searches.

12. For paper selection and data extraction process, the authors should report whether the two reviewers worked independently at each stage of screening and any processes used to resolve disagreements between them.

13. any processes used to obtain or confirm relevant information from study investigators during paper selection and data extraction process?

14. As papers in different languages were included, the authors should report how they translated abstracts or articles into another language to determine their eligibility for paper selection and data extraction.

15. The authors should report how automation tools, such as EndNote, were integrated within the overall study selection process.

16. If any decision rules were used to select data from multiple reports corresponding to a study, and any steps were taken to resolve inconsistencies across reports, the authors should report the rules and steps used.

Study Risk of Bias Assessment:

17. As non-randomized controlled (i.e. experimental) studies were included, the authors should use a more relevant assessment tool, such as RoBANS (Risk of Bias Assessment Tool for Non-Randomized Studies).

Synthesis Methods:

18. The authors should describe the processes used to decide which studies were eligible for each synthesis (both narrative and meta-analysis).

19. The authors should define the comparator, i.e., the control groups, in the eligibility criteria and align it in this section.

20. If sensitivity analyses were performed, the authors should provide details of each analysis, such as the removal of studies at high risk of bias or the use of an alternative meta-analysis model.

Results:

21. Please provide more specific elaboration on the reasons for excluding the 15 papers.

22. Were the design of the two pilot studies the RCTs ? Please confirm whether this review paper only includes RCTs or not. It is necessary to specify the paper selection criteria.

23. Please provide a detailed description of the different intervention components provided by all the included studies.

24. The title on page 12 should be "Study Risk-of-bias Assessment."

26. The risk-of-bias assessment items in the Results section must align with the Method section. For example, the Method section mentions "reporting bias," but it is not mentioned in the Results section. Please also describe in the Method section how to assess "reporting bias;" the authors should check the original SR protocol.

27. Results of individual studies are missing. Please refer to the PRISMA_2020_expanded checklist for details.

Discussion:

28. If a more concrete intervention dosage can be suggested, it would make this paper more meaningful. For example, in line 278, please define "appropriate amounts" of protein and micronutrients more precisely. If no such information can be identified, please explain why and suggest its as the future direction of the study.

29. What is the novelty of this systematic review compared to other similar previous reviews?

30. Overall, the discussion is a bit superficial. It is necessary to be more specific, for example, what type of non-pharmacological interventions are more effective in controlling certain outcomes in people with SO? What possible limitations in the previous studies may affect the results of the findings?

7. PLOS authors have the option to publish the peer review history of their article (what does this mean?). If published, this will include your full peer review and any attached files.

Reviewer #1: No

Reviewer #3: **Yes: **Liu Yat Wa Justina

---

## [Author Response · Author response to Decision Letter 1]

12 Jul 2023

Review Comments to the Author

Reviewer #1: The manuscript was conducted on the impact of non-pharmacological interventions on patients with sarcopenic obesity, indicating that non-pharmacological interventions can effectively improve some clinical symptoms and signs of patients of sarcopenic obesity patients, which is beneficial for clinical promotion and use. The author has made revisions one by one based on the relevant suggestions put forward by the editor. The article has a clear approach and the method is appropriate. It is recommended to publish it.

Response: 

Thank you very much for your comments and support.

Reviewer #3: Thank you for inviting me to review the paper titled "Effects of Non-Pharmacological Interventions on Patients with Sarcopenic Obesity: A Meta-Analysis." It is an interesting review paper, but I have some concerns and suggestions for improvement.

Background

1. The background section needs stronger justifications to support the need for this systematic review and meta-analysis. There have already been several systematic reviews and meta-analyses on this topic, so it is important to address what other researchers have concluded in their papers, what research gaps this paper attempts to address, and why this review is necessary. Commenting on other similar systematic reviews to identify research gaps would strengthen the background section.

Response: 

According to your helpful suggestion, we have revised the background section. 

2. In lines 68-69, the authors state that there is limited literature on the feasibility of the interventions. However, I disagree, as several review papers on interventions for people with sarcopenic obesity have been published, such as “Yin YH, Liu JYW, Fan TM, Leung KM, Ng MW, Tsang TY, Wong KP, Välimäki M. Effectiveness of Nutritional Advice for Community-Dwelling Obese Older Adults With Frailty: A Systematic Review and Meta-Analysis. Front Nutr. 2021 Jun 29;8:619903. doi: 10.3389/fnut.2021.619903.” Thus, the authors should critique other similar systematic reviews to explain why the current review was considered necessary.

Response: 

It has been revised, thanks. 

3. It is unclear whether this paper focuses on all types of non-drug interventions or only exercise and nutritional interventions. The authors should clarify this in the background section and align it with the title, paper selection criteria, and other parts of the manuscript.

Response: 

It has been revised in the section of Introduction.

4.The authors should briefly describe how the interventions (i.e., non-drug, exercise alone, nutrition alone, or combined, etc.) might work on people with sarcopenic obesity to provide a more comprehensive understanding.

Response: 

Thanks for your suggestion, the mechanisms of exercise and nutritional interventions in patients with SO were added in the section of Introduction.

Objective:

5. In line 72, the authors state that the meta-analysis aimed to "confirm" the effects of the interventions…... However, I feel that the word "confirm" is too strong. It would be better to use the word "explore" instead.

Response: 

Thanks for your kindly help, it has been revised.

6. The objective should be revised using the PICO framework or one of its variants to clearly state the comparisons that were made. This will provide more clarity and specificity to the objective statement.

Response: 

We have revised it according to the PICOS principles. 

Methods - paper selection and data extraction process:

7. Please follow the headings and subheadings according to the PRISMA 2020 expanded checklist. For the Methods section, the headings should include eligibility criteria, information sources, search strategy, and selection process, among others, as specified by the checklist. This comment is also relevant to other sections of the manuscript.

Response: 

Thank you very much for your comments, it has been revised. 

8. The authors need to provide a better definition of targeted intervention(s) in the background section and align it with the eligibility criteria in this section.

Response: 

It has been revised.

9. As mentioned in the background section, sarcopenic obesity refers to the coexistence of obesity and sarcopenia. Therefore, the inclusion criteria should include both indicators to reflect this co-existing body structure. This should include at least one criterion for obesity and one for sarcopenia.

Response: 

The diagnostic criteria for sarcopenic obesity was added. 

10. It should specify the year of publication for the studies included in this review paper.

Response: 

It has been revised.

11. It should report whether they conducted any hand searches in addition to their electronic searches.

Response: 

It has been revised.

12. For paper selection and data extraction process, the authors should report whether the two reviewers worked independently at each stage of screening and any processes used to resolve disagreements between them.

Response: 

Related statements have been added. Thank you very much.

13. any processes used to obtain or confirm relevant information from study investigators during paper selection and data extraction process?

Response: 

We can get the needed information from the included literature, so we don't need to seek help from the study investigators. 

14. As papers in different languages were included, the authors should report how they translated abstracts or articles into another language to determine their eligibility for paper selection and data extraction.

Response: 

Thank you very much for your comments. Chinese and English articles were included, articles in other languages were excluded.

15. The authors should report how automation tools, such as EndNote, were integrated within the overall study selection process.

Response: 

Thank you very much for your comments and related statements have been added. 

16.If any decision rules were used to select data from multiple reports corresponding to a study, and any steps were taken to resolve inconsistencies across reports, the authors should report the rules and steps used.

Response: 

We adopt uniform inclusion and exclusion criteria to ensure the consistency of selected studies. 

Study Risk of Bias Assessment:

17. As non-randomized controlled (i.e. experimental) studies were included, the authors should use a more relevant assessment tool, such as RoBANS (Risk of Bias Assessment Tool for Non-Randomized Studies).

Response: 

This meta-analysis only included randomized controlled trials, and no experimental studies were included, so there is no need to use RoBANS tools for evaluation.

Synthesis Methods:

18. The authors should describe the processes used to decide which studies were eligible for each synthesis (both narrative and meta-analysis).

Response: 

The selection process is illustrated in Figure 1 and statemented the section of Material and methods. 

19. The authors should define the comparator, i.e., the control groups, in the eligibility criteria and align it in this section.

Response: 

It has been revised in the section of eligibility criteria. 

20. If sensitivity analyses were performed, the authors should provide details of each analysis, such as the removal of studies at high risk of bias or the use of an alternative meta-analysis model.

Response: 

Sensitivity analyses will performed when P<0.1 and I2≥50%, but the heterogeneity of included studies in this work was less than 50%, so there is no need for sensitivity analysis. 

Results:

21. Please provide more specific elaboration on the reasons for excluding the 15 papers.

Response: 

Reasons for the exclusion of 15 studies have been shown in S5 File. 

22. Were the design of the two pilot studies the RCTs ? Please confirm whether this review paper only includes RCTs or not. It is necessary to specify the paper selection criteria.

Response: 

Thanks, it has been revised. 

23. Please provide a detailed description of the different intervention components provided by all the included studies.

Response: 

It has been revised.

24. The title on page 12 should be "Study Risk-of-bias Assessment."

Response: 

It has been revised. 

In the reviewer's comments, the 25th is of missing. 

26. The risk-of-bias assessment items in the Results section must align with the Method section. For example, the Method section mentions "reporting bias," but it is not mentioned in the Results section. Please also describe in the Method section how to assess "reporting bias;" the authors should check the original SR protocol.

Response: 

It has been added in the section of material and methods. 

27. Results of individual studies are missing. Please refer to the PRISMA_2020_expanded checklist for details.

Response: 

Results of individual studies are shown in Table 1.

Discussion:

28. If a more concrete intervention dosage can be suggested, it would make this paper more meaningful. For example, in line 278, please define "appropriate amounts" of protein and micronutrients more precisely. If no such information can be identified, please explain why and suggest its as the future direction of the study.

Response: 

We have summarized the recommendations for nonpharmacological interventions for patients with SO.

29. What is the novelty of this systematic review compared to other similar previous reviews?

Response: 

The innovation of this meta-analysis is to test whether exercise and nutrition intervention is effective for sarcopenic obesity, and give a reasonable answer to the existing controversy.

30. Overall, the discussion is a bit superficial. It is necessary to be more specific, for example, what type of non-pharmacological interventions are more effective in controlling certain outcomes in people with SO? What possible limitations in the previous studies may affect the results of the findings?

Response: 

Thanks for your comments, and the section of discussion has been revised.

---

## [Editor Report · Decision Letter 2]

2 Aug 2023

Effects of non-pharmacological interventions on patients with sarcopenic obesity: A meta-analysis

PONE-D-23-08981R2

Dear Dr. Chu,

We’re pleased to inform you that your manuscript has been judged scientifically suitable for publication and will be formally accepted for publication once it meets all outstanding technical requirements.

Kind regards,

Meisam Akhlaghdoust, M.D., M.P.H.

Academic Editor

PLOS ONE
---

## [Editor Report · Acceptance letter]

4 Aug 2023

PONE-D-23-08981R2 

Effects of non-pharmacological interventions on patients with sarcopenic obesity: A meta-analysis 

Dear Dr. Chu:

I'm pleased to inform you that your manuscript has been deemed suitable for publication in PLOS ONE. Congratulations! Your manuscript is now with our production department. 

Kind regards, 

on behalf of

Dr. Meisam Akhlaghdoust 

Academic Editor

PLOS ONE